# Differential attainment in specialty training recruitment in the UK: an observational analysis of the impact of psychometric testing assessment in Public Health postgraduate selection

Richard J Pinder ![ORCID],[1,2] Fran Bury ![ORCID],[1] Ganesh Sathyamoorthy,[1,2,3] Azeem Majeed ![ORCID],[1,2] Mala Rao ![ORCID] [1,2,3]

[1]Department of Primary Care and Public Health, School of Public Health, Imperial College London, London, UK
[2]NIHR Northwest London Applied Research Collaboration, Imperial College London, London, UK
[3]Ethnicity and Health Unit, Department of Primary Care and Public Health, School of Public Health, Imperial College London, London, UK

**Correspondence to**
Dr Richard J Pinder;
richard.pinder@imperial.ac.uk

## ABSTRACT

**Objectives** To determine how current psychometric testing approaches used in selection of postgraduate training in UK Public Health are associated with socioeconomic and sociocultural background of applicants (including ethnicity).

**Design** Observational study using contemporaneous data collected during recruitment and psychometric test scores.

**Setting** Assessment centre of UK national Public Health recruitment for postgraduate Public Health training. The assessment centre element of selection comprises three psychometric assessments: Rust Advanced Numerical Reasoning, Watson-Glaser Critical Thinking Assessment II and Public Health situational judgement test.

**Participants** 629 applicants completed the assessment centre in 2021. 219 (34.8%) were UK medical graduates, 73 (11.6%) were international medical graduates and 337 (53.6%) were from backgrounds other than medicine.

**Main outcome measure** Multivariable-adjusted progression statistics in the form of adjusted OR (aOR), accounting for age, sex, ethnicity, professional background and surrogate measures of familial socioeconomic and sociocultural status.

**Results** 357 (56.8%) candidates passed all three psychometric tests. Candidate characteristics negatively associated with progression were black ethnicity (aOR 0.19, 0.08 to 0.44), Asian ethnicity (aOR 0.35, 0.16 to 0.71) and coming from a non-UK medical graduate background (aOR 0.05, 0.03 to 0.12); similar differential attainment was observed in each of the psychometric tests. Even within the UK-trained medical cohort, candidates from white British backgrounds were more likely to progress than those from ethnic minorities (89.2% vs 75.0%, p=0.003).

**Conclusion** Although perceived to mitigate the risks of conscious and unconscious bias in selection to medical postgraduate training, these psychometric tests demonstrate unexplained variation that suggests differential attainment. Other specialties should enhance their data collection to evaluate the impact of differential attainment on current selection processes and take forward opportunities to mitigate differential attainment where possible.

## STRENGTHS AND LIMITATIONS OF THIS STUDY

⇒ This study analyses a complete annual cohort of applicants for UK Public Health specialty training in 2021 to identify the extent and impact of differential attainment arising in the psychometric testing component of selection.

⇒ The methodology employs an enhanced and prospectively collected set of questions that explore sociocultural background of applicants, as well as categorising UK medical graduates, international medical graduates and candidates from backgrounds other than medicine.

⇒ Socioeconomic disadvantage and intersectionality are inherently complex and difficult to quantify; we cannot exclude the possibility of other potential confounders.

## INTRODUCTION

Over recent years, awareness of the challenges facing doctors and other healthcare workers from ethnic minority backgrounds has increased.[1] The inequity of opportunity and outcome is driven by myriad factors, including: barriers to professional progression[2–4]; more active forms of racism from patients, colleagues and institutions[5]; and the interaction with other social categorisations (so-called intersectionality).[6] Differential attainment describes a gap in attainment levels between different demographic groups, including by protected characteristics.[7] Differential attainment has been observed in medical school selection,[8 9] during undergraduate medical training,[10] in postgraduate training[11] in senior appointments[12] and in the field of medical research.[13]

Like other employment sectors, the medical profession internationally has tried to mitigate unconscious bias arising during recruitment.[14] As part of the historical shift from

resource-intensive interviews where unconscious (and potentially even conscious) bias has been suspected,[15] psychometric testing has been commonly perceived as more objective.[2] And while validation of psychometric testing in the context of UK General Practice recruitment[16] and Public Health recruitment[17] has been published, conceptual validity has been largely predicated on subsequent professional progression as opposed to investigating potential differential attainment.

The use and weighting accorded to psychometric tests for postgraduate medical recruitment in the UK varies. The Multi-Specialty Recruitment Assessment[18] (MSRA) is a multi-instrument assessment of procedural and clinical knowledge[19] that began in general practice and which has also been used in ophthalmology and obstetrics and gynaecology for several years.[20] In 2022, around 10 UK specialties used MSRA as a means of shortlisting candidates.

Public Health is an unusual specialty that recruits both medical and non-medical candidates. The latter are referred to as candidates from *backgrounds other than medicine* (BOTM). The current three-stage design of postgraduate specialty recruitment for Public Health (online supplemental figure 1) comprises eligibility checking, an assessment centre (AC) with three psychometric instruments and a selection centre (SC); this process reduces approximately 1000 applicants to 650 applicants at AC and 216 applicants at SC.[17 21] This selection process preceded the introduction of the MSRA.

Due to the diversity of Public Health's applicant pool, MSRA (which assesses elements of clinical decision-making) would not be a fair assessment tool. Instead, since 2009, Public Health has used two generic psychometric instruments that are used internationally in recruiting to corporate roles: Rust Advanced Numerical Reasoning Assessment[22] (RANRA) and Watson-Glaser Critical Thinking Assessment II[23] (WGCTAII). A third bespoke situational judgement test (SJT) instrument is designed and implemented by Work Psychology Group with input from Public Health subject matter experts and was added to the process in 2011. The SJT provides a hypothetical scenario for a Public Health registrar to appraise and then rank in order of suitability their response from a list of stated actions. The three instruments are undertaken as a 170-minute computer-based test. Much-like MSRA, the Public Health AC is operated in multiple locations by the commercial testing company PearsonVue.

In 2020, the BMJ published a news story suggesting that for every specialty included, white candidates were more likely to be deemed appointable for specialty training than ethnic minority candidates.[24] Among the specialties considered, Public Health had the greatest differential between white and ethnic minority candidates in the proportion deemed appointable. Prompted by the article, this research was independently commissioned by the UK Public Health Recruitment Executive Group (a committee of Health Education England) to determine whether some groups of candidates for Public Health specialty training were less likely to be successful in their applications than others, and to determine at which stage(s) of the process this occurs. The analysis, drafting and decision to submit this paper were undertaken exclusively by the authors. This study forms the second and final paper of the commissioned work.

Our preceding paper[21] analysed 2252 applicants over the 3 years (2018–2020) inclusive of the three-stage process. Contrary to initial beliefs that the SC interviews might be the source of the differential attainment, the greatest differential attainment was associated with the psychometric stage of assessment.[21] Applicants from black backgrounds and Asian backgrounds, as well as those from BOTM, were between 70% and 90% less likely to progress from the psychometric testing stage than white British candidates and medical candidates, respectively. As a result of these observations, we considered residual confounding as a possible explanation, recognising our inability to disaggregate international medical graduates (IMGs; who are less likely to be white British) as well as to detect other socioeconomic differences. Also of interest was whether the SJT specifically might be contributing to differential attainment by assuming cultural understanding of the British workplace: for example, in assessing etiquette around punctuality and other culturally nuanced professional behaviours. Finally, we sought to assess whether having a first language other than English affects performance in the time-pressured WGCTAII assessment.

## METHODS

### Data sources

A partially redacted dataset (to comply with the principle of data minimisation[25]) for those applying for training places in 2021 (The year described relates to the August in which successful applicants would take up the role. The recruitment cycle begins the preceding November, such that the year being analysed includes those who began their application in November 2020 and potentially were selected to take up their post in August 2021.) was extracted from the application system ('Oriel') by Health Education England East Midlands. This dataset covered applicant demography (sex, age group and ethnicity), professional background and progression through the selection process. For the first time in 2021, an optional enhanced set of questions were posed to candidates who sought to capture information about sociocultural background (table 1). The data were passed to the research team, who then cleaned and coded the data for analysis.

### Patient and public involvement

No patients involved.

Professional background was assumed based on the eligibility criteria applied by each candidate: whether medical or BOTM. Additionally, we reviewed the degree-awarding institution for all applicants who applied through the medical route to sort this group into UK-trained medical

**Table 1** Enhanced data collection for Public Health Specialty Recruitment, 2021 recruitment round

| Question | Response options | Source/basis of question |
|---|---|---|
| What is the highest level of qualifications achieved by either of your parent(s) or guardian(s) by the time you were 18? | ▶ Degree level or degree equivalent or above (for example, first or higher degrees, postgraduate diplomas, NVQ/SVQ level 4 or 5, etc) <br> ▶ Qualifications below degree level (for example, an A-level, SCE Higher, GCSE, O-level, SCE Standard/Ordinary, NVQ/SVQ, BTEC, etc) <br> ▶ No qualifications <br> ▶ Do not know or cannot remember <br> ▶ Prefer not to say <br> ▶ Not applicable | The question is drawn from UK Government guidance[35] on valid measures of socioeconomic status. More information can be found at: https://assets.publishing.service.gov.uk/government/uploads/system/uploads/attachment_data/file/713739/Annex_A-_Evaluation_of_measures_of_Socio-economic_background.pdf |
| What is your main language? | ▶ English <br> ▶ Welsh <br> ▶ Arabic <br> ▶ Bengali <br> ▶ Chinese—Cantonese <br> ▶ Chinese—Mandarin <br> ▶ French <br> ▶ German <br> ▶ Gujurati <br> ▶ Portuguese <br> ▶ Punjabi <br> ▶ Spanish <br> ▶ Urdu <br> ▶ Other (please complete) | The question is a repeat of the question asked in the UK (national) Census from 2011,[36] which drew on a broad base of stakeholder testing and validation. More information can be found at: https://www.ons.gov.uk/file?uri=/census/2011census/howourcensusworks/howweplannedthe2011census/questionnairedevelopment/finalisingthe2011questionnaire/final-recommended-questions-2011-language_tcm77-183991.pdf |

BTEC, Business and Technology Education Council; GCSE, General Certificate of Secondary Education; NVQ, National Vocational Qualification; SCE, Scottish Certificate of Education; SVQ, Scottish Vocational Qualification.

doctors and those who had trained outside the UK (termed IMGs). Every year, a small number of people with a primary medical qualification apply through the BOTM eligibility criteria. Applicants who hold a primary medical qualification but do not hold UK medical registration and a licence to practise may apply through the BOTM route. It is therefore possible that some doctors, including IMGs, applied through this route, but their identity was not captured from our process, resulting in them being labelled as BOTM.

### Data processing

Data processing and analysis were undertaken in STATA SE V.17.0 for Mac. The primary outcome for the analyses was 'passing AC' defined as having achieved a minimum threshold score in each of the three psychometric assessments. Subanalyses were also undertaken (and are reported) for each psychometric test separately. Due to the comparatively small number of black and minority ethnic candidates, the ethnicity categories were aggregated upwards to create sufficiently large groups for analysis. Responses to the two new optional sociocultural questions were aggregated into: 'parental degree' or 'no parental degree', and 'English as main language' and 'not English as main language'.

### Analytical approach

Descriptive analysis of passing the AC and its constituent parts was undertaken with counts and percentages. Logistic regression was used for both univariable and multivariable analyses producing ORs and adjusted ORs (aORs) with 95% CIs, assuming two-tailed testing with an alpha of 0.05. Applicants for whom demographic data were missing are presented in descriptive analyses with count data but censored in regression analyses. Descriptive analyses where counts are <5 have been masked to reduce the risk of deductive disclosure and maintain privacy.

### RESULTS

There were 963 applicants for Public Health Specialty Recruitment in the 2021 round, of which 629 (65.3%) applicants passed eligibility checking and attended the AC; the results presented are for these candidates only.

Of the candidates attending the AC, approximately two-thirds were female (65.3%), just over half (57.7%) identified themselves as coming from white backgrounds (table 2) and the median age was 32 years. In terms of professional background, 53.6% of AC attendees came from backgrounds other than medicine, while 11.6% and 34.8% were identified as coming from IMG and UK

**Table 2** Descriptive analysis of demographic characteristics of candidates attending assessment centre (AC) 2021, with subanalysis of AC outcome (count and percentage)

| | All candidates sitting AC n (%) | Candidates failing AC n (%) | Candidates passing AC n (%) |
|---|---|---|---|
| **Total (%)** | 629 | 272 (43.2) | 357 (56.8) |
| Sex | | | |
| Male | 199 (31.6) | 89 (32.7) | 110 (30.8) |
| Female | 411 (65.3) | 177 (65.1) | 234 (65.6) |
| Other or not known | 19 (3.0) | 6 (2.2) | 13 (3.6) |
| Ethnic group | | | |
| White British | 304 (48.3) | 98 (36.0) | 206 (57.7) |
| White other | 59 (9.4) | 24 (8.8) | 35 (9.8) |
| Black | 56 (8.9) | 41 (15.1) | 15 (4.2) |
| Asian | 114 (18.1) | 65 (23.9) | 49 (13.7) |
| Mixed | 32 (5.1) | 12 (4.4) | 20 (5.6) |
| Chinese | 11 (1.8) | 7 (2.6) | 4 (1.1) |
| Other | 15 (2.4) | 8 (2.9) | 7 (2.0) |
| Not disclosed | 38 (6.0) | 17 (6.3) | 21 (5.9) |
| Age in years | | | |
| ≤29 | 209 (32.2) | 61 (22.4) | 148 (41.5) |
| 30–34 | 181 (28.8) | 77 (28.3) | 104 (29.1) |
| 35–39 | 96 (15.3) | 53 (19.5) | 43 (12.0) |
| 40–44 | 70 (11.1) | 42 (15.4) | 28 (7.9) |
| 45–49 | 33 (5.3) | 21 (7.7) | 12 (3.4) |
| ≥50 | 9 (1.4) | 5 (1.8) | 4 (1.1) |
| Not known | 31 (4.9) | 13 (4.8) | 18 (5.0) |
| Parent or carer highest qualification | | | |
| Degree | 227 (36.1) | 105 (38.6) | 122 (34.2) |
| No degree | 377 (59.9) | 159 (58.5) | 218 (61.1) |
| Not known | 25 (4.0) | 8 (2.9) | 17 (4.8) |
| Main language | | | |
| English | 449 (71.4) | 183 (67.3) | 266 (74.5) |
| Not English | 31 (4.9) | 18 (6.6) | 13 (3.6) |
| Not known | 149 (23.7) | 71 (26.1) | 78 (21.9) |
| Professional background | | | |
| BOTM | 337 (53.6) | 176 (64.7) | 161 (45.1) |
| IMG | 73 (11.6) | 57 (21.0) | 16 (4.5) |
| UK medical | 219 (34.8) | 39 (14.3) | 180 (50.4) |

AC, Assessment Centre; BOTM, background other than medicine; IMG, international medical graduate.

medical backgrounds, respectively. The response rate to the enhanced question set (for those attending the AC) was 96.0% in relation to parental qualification, but only 76.3% in relation to main language. Of those responding to these optional questions, 37.6% reported their parent or carer holding a degree-level qualification by the time they were 18 years old, and 93.5% reported their main language as being English.

Of those attending the AC, 357 (56.8%) scored sufficiently well across the three assessments to 'pass' it.

Breaking down the applicants by professional background (table 3), female applicants outnumbered males across the groups but particularly so in the BOTM group (75.2% vs 25.0%). The median age was 5 years greater for BOTM candidates than among those from the UK medical graduate group. The BOTM group was more

**Table 3** Descriptive analysis of demographic characteristics of candidates by professional background in assessment centre 2021 (count and percentage)

| | All candidates from backgrounds other than medicine n (%) | All international medical graduate candidates n (%) | All UK medical graduate candidates n (%) |
|---|---|---|---|
| **Total (%)** | 337 | 73 | 219 |
| Sex | | | |
| Male | 82 (24.8) | 31 (43.1) | 86 (41.3) |
| Female | 248 (75.2) | 41 (56.9) | 122 (58.7) |
| Ethnic group | | | |
| White British | 191 (58.6) | <5 | 111 (55.0) |
| White other | 30 (9.2) | 13 (20.0) | 16 (8.0) |
| Black | 24 (7.4) | 23 (35.4) | 9 (4.5) |
| Asian | 52 (16.0) | 18 (27.7) | 44 (22.0) |
| Mixed | 17 (5.2) | 5 (7.7) | 10 (5.0) |
| Chinese | 5 (1.5) | <5 | 5 (2.5) |
| Other | 7 (2.2) | <5 | 5 (2.5) |
| Median age in years (IQR) | 34 (31–41) | 31 (29–36) | 29 (27–31) |
| Parent or carer highest qualification | | | |
| Degree | 192 (58.4) | 50 (72.5) | 135 (65.5) |
| No degree | 137 (41.6) | 19 (27.5) | 71 (34.5) |
| Main language | | | |
| English | 252 (95.8) | 41 (71.9) | 156 (97.5) |
| Not English | 11 (4.2) | 16 (28.1) | <5 |

IQR, Interquartile range.

socioeconomically diverse than the two medical groupings with 41.6% (of responding applicants) reporting a parent or carer without a degree.

Logistic regression analyses (table 4) suggested that sex, age and main language were not statistically associated with passing the AC (termed hereafter 'progression'). However, there was statistically significant lower progression among applicants from black backgrounds (aOR 0.19, 95% CI 0.07 to 0.47) and Asian backgrounds (aOR 0.36, 95% CI 0.19 to 0.66). The association of lower progression probability with advancing age was not statistically significant after adjustment.

Professional background was also strongly associated with progression: compared with UK medical graduates, IMGs were more than 90% less likely to progress (aOR 0.07, 95% CI 0.03 to 0.17) and BOTM candidates approximately 80% less likely (aOR 0.21, 95% CI 0.11 to 0.39). Subgroup analysis of UK medical graduates only, where ethnicity was recorded (n=200), showed a 14.2% percentage point gap between white British candidates (89.2% progressed) and those who were not white British (75.0%, p=0.003).

Descriptive and inferential analyses of the RANRA, Watson-Glaser and SJT instruments separately demonstrated broadly similar results (online supplemental tables 1–3). The highest pass rate (79.3%) was achieved on the RANRA (numerical reasoning) assessment, followed by Watson-Glaser (74.1%) then SJT (73.9%).

Candidates from black backgrounds (black African, black Caribbean and black other) returned the lowest pass rates across all three instruments, followed by those from Asian backgrounds (Indian, Pakistani, Bangladeshi and other Asian). The performance gap by those applicants from black backgrounds was smallest on the SJT. The 'other' ethnic group also returned lower scores in univariable regression but after adjustment, this was not the case. Younger age was associated with higher probability of passing—but the association was attenuated to a borderline finding following adjustment. UK medical graduates were significantly more likely to pass than the other two professional groups, with IMGs receiving the lowest scores overall. UK medical graduates showed the greatest performance gap with BOTM applicants in the RANRA (95.4% passing vs 74.5%) and the SJT (93.2% passing vs 67.4%).

Parental qualification was broadly not associated with performance (except weakly following adjustment in Watson-Glaser). Univariable regression suggested that main language other than English might be associated with lower scores across instruments; however, these

**Table 4** Univariable and multivariable logistic regression of demographic associations of candidates passing AC (ORs, adjusted ORs, 95% CIs and p values), with subgroup analysis of UK-trained medical applicants (count, percentage and $\chi^2$ p value)

| | OR (95% CI) | P value | Adjusted OR* (95% CI) | P value |
|---|---|---|---|---|
| **Sex** | | | | |
| Male | Ref | | Ref | |
| Female | 1.07 (0.76 to 1.50) | 0.70 | 1.20 (0.73 to 1.97) | 0.46 |
| **Ethnic group** | | | | |
| White British | Ref | | Ref | |
| White other | 0.69 (0.39 to 1.23) | 0.21 | 1.10 (0.50 to 2.46) | 0.89 |
| Black | 0.17 (0.09 to 0.33) | <0.001 | 0.19 (0.07 to 0.47) | <0.001 |
| Asian | 0.36 (0.23 to 0.56) | <0.001 | 0.36 (0.19 to 0.66) | <0.001 |
| Mixed | 0.79 (0.37 to 1.69) | 0.55 | 1.35 (0.49 to 3.69) | 0.56 |
| Chinese | 0.27 (0.08 to 0.95) | 0.04 | 0.20 (0.03 to 1.58) | 0.13 |
| Other | 0.42 (0.15 to 1.18) | 0.10 | 0.45 (0.11 to 1.80) | 0.26 |
| **Age in years†** | 0.92 (0.90 to 0.95) | <0.001 | 0.96 (0.93 to 1.00) | 0.07 |
| **Parent or carer highest qualification** | | | | |
| No degree | Ref | | Ref | |
| Degree | 1.18 (0.85 to 1.64) | 0.32 | 1.55 (0.99 to 2.43) | 0.06 |
| **Main language** | | | | |
| English | Ref | | Ref | |
| Not English | 0.50 (0.34 to 1.04) | 0.06 | 1.37 (0.52 to 3.67) | 0.52 |
| **Professional background** | | | | |
| BOTM | 0.20 (0.13 to 0.30) | <0.001 | 0.21 (0.11 to 0.39) | <0.001 |
| IMG | 0.06 (0.03 to 0.12) | <0.001 | 0.07 (0.03 to 0.17) | <0.001 |
| UK medical | Ref | | Ref | |
| **Subgroup analysis of UK-trained medical applicants only** | **Passed AC n** | **(%)** | **$\chi^2$ p value** | |
| White British (n=111) | 99 | (89.2) | 0.003 | |
| Ethnic minority (n=89) | 81 | (75.0) | | |

*Adjustment includes sex, ethnic group, age, parent or carer degree status, first language and professional background.
†Age as a continuous variable in years.
AC, assessment centre; BOTM, backgrounds other than medicine; IMG, international medical graduate.

associations were attenuated following adjustment in the multivariable regression.

## DISCUSSION
### Statement of principal findings
Coming from a black ethnic group, Asian ethnic group or from a non-UK medical graduate background were all independently predictive of poorer performance across all three elements of the UK psychometric tests used in Public Health Specialty Recruitment in 2021. We detected no consistent pattern of lower-scoring performance associated with English as a second language, or from socioeconomic background (as measured by parental qualification). Increasing age may be associated with poorer performance, but some of the previously noted performance gap appeared to be accounted for by confounding (possibly due to medical applicants, who

perform better overall, tending to be younger). Yet, even within the UK-trained medical cohort of applicants, there was differential attainment by ethnicity.[3]

The hypothesis, outlined in our previous paper,[21] that sociocultural factors (including first language) might be causing poorer performance in the SJT than the highly time-pressured Watson-Glaser, was not supported by the analysis. We had also hypothesised that the use of SJTs in other medical application processes might have conferred an advantage to UK medical graduates, and language fluency has previously been associated with SJT performance in the recruitment process for general practice.[3] Yet, the performance difference in the SJT was not substantially different to the gap observed in the RANRA. Instead, we observed consistent performance across all three instruments, with significantly lower performance by candidates from black and Asian backgrounds. Black

candidates were 80% less likely than white British applicants to pass the AC in 2021, and Asian candidates 60% less likely.

That these differences persisted even after adjustment for professional background and subcategorisation of IMGs is deeply concerning. A further striking finding is that UK medical graduates were 10 times more likely to pass the AC than IMGs. Once again, this difference persisted after multivariable adjustment.

Each of these three psychometric tests was developed and validated independently. Each test is designed to test a separate aspect of potential suitability for Public Health training. Yet, similar patterns of differential attainment are observed across all three. Together, these findings add to the literature that suggests differential attainment is present across a wide range of psychometric tests.[26 27]

The earlier working hypothesis that age was likely confounded by professional background appears broadly justified although there remains some residual association after adjustment. The wider distribution of ages in the BOTM group may explain some of this difference. Although the two medical groups were socioeconomically more advantaged overall than the BOTM group, it was notable that the socioeconomic metric was not strongly associated with progression. However, the cohort entering the process appears more homogeneously drawn from higher socioeconomic groups than the UK population average. Accordingly, a type II error cannot be ruled out.

### Strengths and weaknesses of the study

To our knowledge, this is the first study to systematically analyse differential attainment in a UK national selection process: while other analyses have made high-level comparisons, this study has examined in detail the intricacies of a single large-scale selection process and sought to anticipate confounding through the use of bespoke sociocultural questions, prospectively integrated into the process. Yet, residual confounding cannot be ruled out: the sociocultural variables used in the regression are imperfect, and it is very possible that other confounding variables were in play. Our findings have important implications for both Public Health as a specialty in the UK, and for the many other medical specialties that employ psychometric testing in their selection processes.

Among the study's weaknesses is that ethnic groupings had to be aggregated to mitigate the analytical risk arising from small numbers. Accordingly, type I and type II errors cannot be ruled out. However, the consistency and strength of the findings around ethnicity and professional background suggest that a type I error is highly unlikely in this context. The single-year nature of the dataset is a limitation, but for those variables measured since 2018,[21] the associations are consistent over 4 years of recruitment: we note the proportion of candidates passing the AC in this year (56.8%) is very similar to the preceding year's cohort (56.5%). It is not possible to infer what effect the COVID-19 pandemic might have had (with the AC taking place in February 2021 and most candidates completing the tests at home rather than in a commercial test centre).

The application of the sociocultural questions, while drawn from best available evidence on the measurement of socioeconomic status, may have been insufficient to detect differences, and the concept of socioeconomic status is complex. We note that 'first language' may not be easy to define for those who have grown up bilingual. The response rate on parental qualification (at 96.0%) was very high; for the language question, response rate was lower at 76.3%. The reason for the differential response rate on two sequential questions is unclear.

### Strengths and weaknesses in other studies, discussing important differences in results

As already noted, in the context of the published literature, these differential attainment gaps are not as surprising as they might at first appear.[2] There is already extensive evidence of differential attainment relating to ethnicity across medical training in the UK.[28] Likewise, the broader evidence base on psychometric testing shows relatively consistent findings that older and ethnic minority candidates generally perform less well on standardised testing.[26] While there remains some uncertainty over the precise drivers, possible explanations include specific barriers, such as differential access to networks, familiarity with psychometric testing,[29] test taker perception,[26] stereotype threat,[30] through to systemic disadvantage and structural racism occurring through the life-course. Such putative explanatory variables are extremely difficult to measure or indeed analyse in a quantitative methodology.

### Meaning of the study: possible explanations and implications for clinicians and policymakers

Due to Public Health recruitment drawing on a more diverse candidate pool than most other medical specialties (that require a primary medical qualification), comparing Public Health recruitment with other national programmes is difficult. Successful completion of medical school and foundation training is likely to create a more homogeneous and high-performing cohort of applicants. However, this study suggests that the observed differential attainment cannot be attributed to confounding alone: likely differential attainment was observed even within the UK medical graduate group. It appears that the system is unfairly discriminating against certain candidates.

While the status quo is unacceptable, we recognise the validity of the current recruitment process which has demonstrated that the candidates selected in the current process progress well in later Public Health training.[17] However, there is some potential circularity to such a validation: that by selecting out people (through differential attainment) before training scheme entry, the trainees who do proceed are less subject to differential attainment later, leading to the appearance of a valid process. Differential attainment has been seen in postgraduate training in other specialties.[11] Any policy responses to this analysis need to preserve the strengths of the current system.

Reducing a cohort of applicants from approximately 1000 to 100 needs a scalable, objective and practicable approach, and there is little doubt that the current system achieves some, if not most, of those objectives, at least in part.

In our preceding paper,[21] we outlined some of the potential responses to apparent differential attainment in the recruitment process more generally. In the context of these findings from the AC stage specifically, the potential response options include a comprehensive review of the psychometric tests used to devise an assessment process and additional support for candidates from disadvantaged groups.

The implications of these findings reach well beyond Public Health. While most national recruitment processes draw only on medical graduates, the apparent effect of ethnicity among UK medical graduates, coupled with the apparent gap among IMGs, has important ramifications for all medical selection processes that are using psychometric tests. We propose that the assumption for selector teams must now be that there is differential attainment in all psychometric test instruments until proven otherwise. We encourage other specialties to explore the data on their own processes and to integrate prospective monitoring processes such as the sociocultural questions that we have employed in Public Health. Such efforts are imperative if we are to mitigate the risk of group-think perpetuating the selection of people who think like the 'establishment'.

### Unanswered questions and future research

This study has identified likely differential attainment in the current national selection process for Public Health. And while it increasingly appears that differential attainment is a feature of psychometric testing in its present form, we are unable to clearly recommend mitigation that would be both simple and uncontentious. While there is a need for additional research, as a medical profession, we must be careful to avoid additional analysis becoming a rationale for delaying our response.

A range of additional variables may be useful. Neurodiversity was not included in the analysis although candidates declaring such conditions were provided with reasonable adjustments which included additional time and/or adjustments to the examination logistics. Previous work among trainees in general practice has observed higher prevalence of dyslexia among IMGs who had failed their first applied knowledge test.[31] Likewise, prior education or educational performance was also not included in the analysis. Previous work has suggested possible intersectional associations with ethnicity in respect of prior education.[32] Capturing such data (while acknowledging the arguably wider range of educational backgrounds from which Public Health applicants come) is an area for potential further work.

There is also a more fundamental limitation of quantitative approaches to 'measuring' diversity[33]: the quantitative paradigm is problematic in recognising intersectionality as well as other contributing factors that motivate fairer selection. Some psychometricians have also challenged the assumption (the so-called Cultural Test Bias Hypothesis) that the observation of mean differences can be inferred as bias.[34] But in the absence of alternatives and the sheer volume of applicants and the differences observed, this analysis is an important start.

### CONCLUSIONS

All doctors and Public Health professionals need to have confidence in national systems of selection. We need a senior medical and public health workforce that better reflects the populations from which it is drawn and which they seek to serve. The drivers of differential attainment are complex, and policy responses even more so.[2 33] However, Public Health is a specialty that values diversity and promotes equity of opportunity. It is vital that we create a fairer future for doctors, healthcare and policy professionals.

**Contributors** RJP and FB conceived the study and undertook the analyses that are presented in this paper. RJP drafted the original version of this manuscript and amendments following peer review. FB, GS, AM and MR reviewed and revised the draft. MR is the guarantor of this paper.

**Funding** Imperial College London is grateful for support from the NW London NIHR Applied Research Collaboration.

**Disclaimer** The views expressed in this publication are those of the authors and not necessarily those of the NIHR or the Department of Health and Social Care.

**Competing interests** RJP is a fellow of the Faculty of Public Health and former technical assessment lead within the national Public Health Recruitment Executive Group hosted by Health Education England. FB is a member of the Faculty of Public Health and a member of the Faculty's Special Interest Group in Equality and Diversity. FB is also a specialty registrar in Public Health. AM is a fellow of the Faculty of Public Health. MR is a fellow of the Faculty of Public Health and former medical advisor to NHS England on Workforce Race Equality.

**Patient and public involvement** Patients and/or the public were not involved in the design, or conduct, or reporting, or dissemination plans of this research.

**Patient consent for publication** Not required.

**Ethics approval** The analyses are evaluation and the data were taken from an administrative dataset. Permission was granted by the data owner to use these data for the explicit analytical purposes of this paper, but these analyses and approach were not anticipated at the time that the data were collected.

**Provenance and peer review** Not commissioned; externally peer reviewed.

**Data availability statement** Data may be obtained from a third party and are not publicly available.

## ORCID iDs
Richard J Pinder http://orcid.org/0000-0002-7010-6009
Fran Bury http://orcid.org/0000-0002-6305-1994
Azeem Majeed http://orcid.org/0000-0002-2357-9858
Mala Rao http://orcid.org/0000-0001-5504-6303

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
