## [Reviewer comments · BMJ Open]

ARTICLE DETAILS

TITLE (PROVISIONAL)	Differential attainment in specialty training recruitment in the United Kingdom: an observational analysis of the impact of psychometric testing assessment in Public Health postgraduate selection.
AUTHORS	Pinder, Richard; Bury, Fran; Sathyamoorthy, Ganesh; Majeed, Azeem; Rao, Mala

VERSION 1 – REVIEW

REVIEWER	Siriwardena, Aloysius University of Lincoln, Community and Health Research Unit
REVIEW RETURNED	17-Dec-2022

GENERAL COMMENTS	Thank you for asking me to review this paper. The inclusion of parental socioeconomic status and first language of candidates in the analysis is novel. The general conclusion that psychometric tests need to be investigated for differential attainment and the reasons for these explored is supported by the study and that an alternative valid and fair assessment should be considered is a fair implication. The problem of unknown confounders is listed in the limitations but even known confounders are not accounted for (see below). The discussion then ignores this gap in the analysis and states that differential attainment is 'unfairly discriminating against certain candidates'. Assertions in the paper, in particular, that the results mean unfair discrimination against minority ethnic candidates, and the solutions presented, for example to set different thresholds for different groups of candidates are inconsistent with the analysis and other arguments presented in the paper. There are limitations which are not discussed. One potential confounder is specific learning difficulty which can affect performance, particularly where this is not declared or where reasonable accommodations are not provided (see Postgraduate Medical Journal 2018; 94:198–203. DOI:10.1136/postgradmedj-2017-135326.). Another important confounder, that is not included in the analysis, is prior education which is known to be linked to performance in undergraduate and postgraduate exams in medicine (BMC Med 2013;11:242. doi: 10.1186/1741-7015-11-242). This will be important for international medical graduates and but may be a factor even in UK graduates, whether medical or non-medical. As acknowledged by the authors, complex sociocultural factors affecting education may also intersect with ethnicity in ways that are not accounted for by a broad measure such as socioeconomic status.
---

	The findings of differential attainment do support, as stated in the discussion, 'review of the psychometric tests used' and 'additional support for candidates from disadvantaged groups' but it is inconsistent to argue for 'different thresholds set for different groups of candidates' at the same time as arguing that these assessments are not valid. It is difficult to see on what basis this could be done, and no evidence is provided that this could work effectively or be deemed acceptable to candidates or those selecting them. The authors state that 'observed differential attainment cannot be attributed to confounding alone: differential attainment was observed even within the UK medical graduate group. Instead, it appears that the system is unfairly discriminating against certain candidates.' This is at odds with the statement in the 'Strengths and limitations of the study' that 'Socioeconomic disadvantage and intersectionality are inherently complex and difficult to quantify; we cannot exclude the possibility of other potential confounders.' Some of the 'precise drivers' or 'possible explanations' such as test familiarity, are mentioned in the discussion but not accounted for in the analysis, but other known confounders such as educational background and specific learning difficulty are not mentioned. The statement in the abstract that 'psychometric tests demonstrate unwarranted and unacceptable differential attainment' is therefore not supported. Rather, differential attainment is unexplained by the factors accounted for in the analysis presented and how these were operationalised.
--	--

REVIEWER	Patterson, Fiona University of Cambridge, Psychology
REVIEW RETURNED	19-Dec-2022

GENERAL COMMENTS	This paper addresses a very important research area regarding differential attainment (DA) in selection practices. It is a generally well-written paper with a well-executed analysis. However, my main reservation (as the authors acknowledge themselves), is that the material covered is a (relatively small) single specialty in a single recruitment round, based only in the UK. Whilst the findings will be of interest to policymakers involved in UK Public Health specialty recruitment (i.e., identified DA), I do not believe this paper currently tells us anything new regarding DA and selection methods/psychometric testing. Therefore, this paper in its current form is unlikely to be of sufficient interest to the broad readership of the journal/relevance internationally. This is not to say that the topic of DA and selection practices does not have relevance more generally, rather I'm not sure what this paper adds to what is already known regarding DA (and psychometric testing specifically). There are many existing large-scale studies identifying and exploring DA in selection and assessment (including meta-analytic studies within medical education e.g., Woolf et al, 2013 and more recent meta-analytic studies in the selection research and psychometric testing literature covering a broad range of occupational groups, e.g., Sackett, et al, 2022). Several existing studies offer a deeper analysis and insight into the issues regarding DA beyond this current paper. We have known for some while that DA exists in selection practices, the key challenge for researchers is to arrive at a better (causal) explanation as to why such differences exist, to inform more targeted
--

	interventions and policy initiatives to be of interest to the broader academic community. Although this paper is generally well-written, my advice would be for the authors to extend the current analysis to explore causal factors and psychometric test/selection method/selection process design interventions that have addressed the DA issue. Originality - does the work add enough to what is already in the published literature? If so, what does it add? Please cite relevant references to support your comments on originality. Being aware of, and monitoring, DA in medical education and psychometric testing is a very important issue. The authors provide a more granular look into individuals elements of the UK Public Health assessment process to identify potential sources of sub-group differences. However, this article does not add much beyond what is already present in current literature, especially within the selection literature (e.g. Bury et al., 2022; Fyfe et al., 2021; Patterson et al, 2016; Kelly et al., 2021; Sackett et al., 2021). There is a substantial evidence base which suggests that group differences are present across almost all selection methods (including psychometric tests). Examples include CVs (Derous & Ryan 2018), cognitive ability tests (Sackett et al., 2021; Ones et al., 2017) and interviews (Tridente et al., 2022). Although this paper does provide a granular look into individual elements of this specific and unique assessment process to identify DA there is insufficient material as it stands to demonstrate originality. In order to demonstrate originality I would encourage the authors to look at the broader literature as the starting point and then present data and analysis as case material which takes the debate to the next level in examining potential casual factors (to include an analysis of the selection criteria, scoring approaches and the interview etc, beyond the psychometric tests alone and follow-up interventions put in place that aim to reduce DA). A more holistic approach (which is where the literature in this area is heading) to include how DA has been addressed in the selection system would be of much broader interest in dealing with the complexity of the issues. Importance of the work to general readers - does this work matter to clinicians, researchers, policymakers, educators, or patients? Will it help our readers to make better decisions and, if so, how? Is a general medical journal the right place for it? Identifying the presence of DA in psychometric testing is of limited interest to researchers, since the paper does not make any alternative suggestions for methods of assessment, demonstrate what alternative interventions work (or not), such that practically it will not help readers to make changes or better decisions. The article may be better placed in a Public Health Journal as the topic is specific to Public Health and selection methods within that, which as the authors describe, is an “unusual specialty that recruits both medical and non-medical candidates”. Therefore, it may not be of interest to the wider readership as it is not directly transferable to other specialties or the use of these selection tools in other contexts. The paper is based on UK case material that may not be of huge relevance to an international audience, particularly those where their recruitment practices differ greatly. If the case is to be made regarding DA and psychometric testing more broadly, then drawing upon international research on this topic would help improve the importance of the work to the general readership. Is the research question clearly defined and appropriately answered? The research questions are generally well-defined although they could be stated more clearly. The authors state that following the
--	---

	findings from their previous study they are looking at how cultural factors could influence SJT scores, and language could influence scores of the other elements of the assessment centre. Cultural factors and what is included within that analysis would benefit from clearer definition. Overall design of study - appropriate and adequate to answer the research question? The authors explain their reasoning for only looking at the psychometric element of the Public Health recruitment process however, it is important to look at all aspects of the selection process together for this cohort in order to have an accurate picture of the contribution of each element to the DA they are observing in their data. The authors argue that their previous paper has identified a problem with the psychometric element of the process however, this analysis was run on previous cohorts (2018-2020), and we cannot assume that the results will match those from the 2021 cohort. Research suggests using multiple selection tools and using a compensatory method improves group differences outcomes (e.g., Stegers-Jager, 2017). I think the paper would be greatly improved if the authors included all stages of the recruitment process as more comprehensive information/measures in their analysis. This would greatly help the broader discussion, especially in exploring the causal factors at play. Participants - adequately described, their conditions defined, inclusion and exclusion criteria described? How representative were the authors of patients whom this evidence might affect? The target sample has been clearly stated by the authors however, the inclusion criteria could be described with more detail. For example, the authors state that the dataset was partially redacted but do not clarify on what basis. Authors could consider restructuring this section to better the flow of the article and increase reader understanding. Furthermore, the authors may also want to consider/acknowledge whether the 2021 data may have been affected by the COVID-19 pandemic, which has been apparent in other recent selection research studies. Methods - adequately described? Main outcome measure clear? Is the study fully reported in line with the appropriate reporting statement or checklist? Was the study ethical (this may go beyond simply whether the study was approved by an ethics committee or IRB)? The methods section and statistical analysis section is generally clear and coherent although a little vague in parts (for example, it is mentioned that the data was cleaned and coded, and it would be useful to provide an explanation as to how this was done). The research method and statistical analysis are appropriate to answer the research question and the outcome measure is clearly stated. There was some consideration for sample sizes when creating ethnic subcategories. The authors attempt to group ethnicities into categories that will allow for meaningful interpretation while also maintaining as much granularity. However, the sample sizes for some of the ethnic subgroups is still small, and there are large differences in size between the groups, leading the reader to question whether meaningful conclusions can be drawn from this. Results - answer the research question? Credible? Well presented? I am not fully familiar with the methodology deployed but the choice of analysis seems appropriate, and the results have been adequately presented and interpreted. Interpretation and conclusions - warranted by and sufficiently derived from/focused on the data? Discussed in the light of previous evidence? Is the message clear?
--	--

	The authors of the article reach a largely appropriate conclusion and interpretations of the findings with a clear message that the current psychometric tests used in the selection process shows some evidence of DA and needs to be reviewed. There is limited reference to previous research as a way of backing up claims and interpretations of the findings in the discussion. The authors should also work on placing these findings into the wider context of DA in selection (as suggested above). The discussion does not currently sufficiently acknowledge the complexity of the issues in that DA is a complex, multi-faceted issue to which there is no single simple solution (see Woolf, 2020; Clapp & Gordon, 2021). The interpretation seems stops short as it assumes the group differences which are observed are equivalent to negative bias, which is not necessarily the case, which needs to be discussed (Davis et al., 2013; Reynolds, Altmann & Allen, 2021). The study has also been conducted on a cohort (2021) that was affected by Covid. Therefore, there are a number of potentially confounding variables that might have influenced the results which have not been considered such as test-taking environment and access to resources. Factors such as the weighting of the tests may also have had an impact on the group difference outcomes observed in practice (Lievens, Sackett & De Corte, 2021). Ensuring the person specifications and test specification is regularly reviewed and updated is also very important so that they remain relevant to the role, especially given changes to working in a world post-Covid. The authors stress the relevance of the results to policymakers and other specialties, however, make no attempt to suggest ways to mitigate the group differences that are observed or outline any practical actions that could be taken. Further discussion here would greatly improve the paper. In the introduction the authors reference the UK Multi-specialty recruitment assessment (MSRA) and draw comparisons between the Public Health recruitment process and the MSRA. The MSRA is a very different selection test to those used in UK Public Health (which are cognitive ability tests and an SJT that assumes no clinical knowledge), as the MSRA is a test of procedural and clinical knowledge (see Lievens & Patterson, 2011). There also exists a substantial body of work exploring DA in the MSRA (e.g., see Patterson et al, 2018) to identify causal factors which need to be referenced. I think the authors would be better served to make more general comments about psychometric tests used for screening/short-listing purposes, to also be of interest to a broader audience. References - up to date and relevant? Any glaring omissions? The references included are largely relevant and up to date. However, more research could be included as the authors currently do not consider important (recent and more broad ranging) references such as Sackett et al, 2022. Some of the latest research on the topic of DA in the last couple of years, particularly in a healthcare/assessment context, should also be included. Abstract/summary/key messages/what this paper adds - reflect accurately what the paper says? While the conclusions drawn lack some important considerations (as above), the abstract, summary and key messages are reflective of the content of the paper. References Used Fran Bury, Richard J Pinder, & Richard Pinder. (2022). Differential attainment in public health specialty training recruitment in the United Kingdom: an observational analysis of applicants from 2018 to 2020. Journal of Public Health. https://doi.org/10.1093/pubmed/fdac122
--	--

Clapp, J. T., & Gordon, E. K. (2021). Selecting trainees: Too much focus on predictive metrics, not enough on holistic review. *Medical Education*, 56(2), 139–141. <https://doi.org/10.1111/medu.14704>

Davis, D., Dorsey, J. K., Franks, R. D., Sackett, P. R., Searcy, C. A., & Zhao, X. (2013). Do Racial and Ethnic Group Differences in Performance on the MCAT Exam Reflect Test Bias? *Academic Medicine*, 88(5), 593–602. <https://doi.org/10.1097/acm.0b013e318286803a>

Derous, E., & Ryan, A. M. (2018). When your resume is (not) turning you down: Modelling ethnic bias in resume screening. *Human Resource Management Journal*, 29(2), 113–130. <https://doi.org/10.1111/1748-8583.12217>

Fyfe, M., Horsburgh, J., Blitz, J., Chiavaroli, N., Kumar, S., & Cleland, J. (2021). The do's, don'ts and don't knows of redressing differential attainment related to race/ethnicity in medical schools. *Perspectives on Medical Education*, 11(1), 1–14. <https://doi.org/10.1007/s40037-021-00696-3>

Kelly, L., & Sankaranarayanan, S. (2021). Differential attainment: how can we close the gap in paediatrics? *Archives of Disease in Childhood - Education & Practice Edition*, edpract-2020. <https://doi.org/10.1136/archdischild-2020-321066>

Lievens, F., & Patterson, F. (2011). The validity and incremental validity of knowledge tests, low-fidelity simulations, and high-fidelity simulations for predicting job performance in advanced-level high-stakes selection. *Journal of Applied Psychology*, 96(5), 927–940. <https://doi.org/10.1037/a0023496>

Lievens, F., Sackett, P. R., & De Corte, W. (2021). Weighting admission scores to balance predictiveness-diversity: The Pareto-optimization approach. *Medical Education*, 56(2), 151–158. <https://doi.org/10.1111/medu.14606>

Ones, D. S. S. D. (2017, March 27). *Cognitive Ability | 11 | v2 | Measurement and Validity for Employee Se.* Taylor & Francis. Retrieved October 7, 2022, from <https://www.taylorfrancis.com/chapters/edit/10.4324/9781315690193-11/cognitive-ability-deniz-ones-stephan-dilchert-chockalingam-viswesvaran-jes%C3%BAs-salgado>

Patterson, F., Zibarras, L., & Ashworth, V. (2016). Situational judgement tests in medical education and training: Research, theory and practice: AMEE Guide No. 100. *Medical Teacher*, 38(1), 3–17. <https://doi.org/10.3109/0142159x.2015.1072619>

Patterson F, Tiffin P, Lopes S & Zibarras L. (2018). Unpacking the 'dark variance' of differential attainment in professional exams for overseas graduates. *Medical Education*, 52(7)., DOI:10.1111/medu.13605

Reynolds, C.R., Altmann, R.A., Allen, D.N. (2021). The Problem of Bias in Psychological Assessment. In: *Mastering Modern Psychological Testing*. Springer, Cham. https://doi.org/10.1007/978-3-030-59455-8_15

Stegers-Jager, K. M. (2017b). Lessons learned from 15 years of non-grades-based selection for medical school. *Medical Education*, 52(1), 86–95. <https://doi.org/10.1111/medu.13462>

Sackett, P et al (2022). Revisiting meta-analytic estimates of validity in personnel selection: Addressing systematic overcorrection for restriction of range. *Journal of Applied Psychology*, 107(11):2040-2068. doi: 10.1037/apl0000994.

Tridente, A., Parry-Jones, J., Chandrashekaraiyah, S., & Bryden, D. (2022). Differential attainment and recruitment to Intensive Care Medicine Training in the UK, 2018–2020. *BMC Medical Education*, 22(1). <https://doi.org/10.1186/s12909-022-03732-w>

	Woolf, K. (2020). Differential attainment in medical education and training. BMJ , m339. https://doi.org/10.1136/bmj.m339
--	---

VERSION 1 – AUTHOR RESPONSE

	Reviewer 1 comment (AS)	Response
1	Thank you for asking me to review this paper. The inclusion of parental socioeconomic status and first language of candidates in the analysis is novel. The general conclusion that psychometric tests need to be investigated for differential attainment and the reasons for these explored is supported by the study and that an alternative valid and fair assessment should be considered is a fair implication.	Thank you for your comments – these are very helpful and we hope you will agree with us that the addition of these further limitations and citation of the papers mentioned improves the quality of the paper. We have provided a point-by-point response below.
2	The problem of unknown confounders is listed in the limitations but even known confounders are not accounted for (see below). The discussion then ignores this gap in the analysis and states that differential attainment is 'unfairly discriminating against certain candidates'.	We accept this point and have sought to qualify our conclusions. We have replaced the phrase “this study confirms” with “this study suggests” and we have also qualified later in the sentence that “likely” differential attainment was observed. In response to the citation in the reviewer response, we note that it was preceded with “it appears”: However, this study suggests that the observed differential attainment cannot be attributed to confounding alone: likely differential attainment was observed even within the UK medical graduate group. Instead, it appears that the system is unfairly discriminating against certain candidates. We have added the following into the first paragraph under “Strengths and weaknesses of the study”: Yet residual confounding cannot be ruled-out: the sociocultural variables used in the regression are imperfect, and it is very possible that other confounding variables were in-play.
3	Assertions in the paper, in particularly, that the results mean unfair discrimination against minority ethnic candidates, and the solutions presented, for example to set different thresholds for different groups of candidates are inconsistent with the analysis and other arguments presented in the paper.	See response to (6)

4	There are limitations which are not discussed. One potential confounder is specific learning difficulty which can affect performance, particularly where this is not declared or where reasonable accommodations are not provided (see Postgraduate Medical Journal 2018; 94:198–203. DOI:10.1136/postgradmedj-2017-135326.).	Thank you – this is a really helpful suggestion and we have added this to the “Unanswered questions and future research”: However, a range of additional variables may be useful. Neurodiversity was not included in the analysis although candidates declaring such conditions were provided with reasonable adjustments which included additional time and / or adjustments to the exam logistics. Previous work among trainees in general practice has observed higher prevalence of dyslexia among international medical graduates who had failed their first applied knowledge test (AKT).²⁹ Likewise, prior education or educational performance was also not included in the analysis. Previous work has suggested possible intersectional associations with ethnicity in respect of prior education.³⁰ Capturing such data (while acknowledging the arguably wider range of educational backgrounds from which Public Health applicants come) is an area for potential further work.
5	Another important confounder, that is not included in the analysis, is prior education which is known to be linked to performance in undergraduate and postgraduate exams in medicine (BMC Med 2013;11:242. doi: 10.1186/1741-7015-11-242). This will be important for international medical graduates and but may be a factor even in UK graduates, whether medical or non-medical. As acknowledged by the authors, complex sociocultural factors affecting education may also intersect with ethnicity in ways that are not accounted for by a broad measure such as socioeconomic status.	However, a range of additional variables may be useful. Neurodiversity was not included in the analysis although candidates declaring such conditions were provided with reasonable adjustments which included additional time and / or adjustments to the exam logistics. Previous work among trainees in general practice has observed higher prevalence of dyslexia among international medical graduates who had failed their first applied knowledge test (AKT).²⁹ Likewise, prior education or educational performance was also not included in the analysis. Previous work has suggested possible intersectional associations with ethnicity in respect of prior education.³⁰ Capturing such data (while acknowledging the arguably wider range of educational backgrounds from which Public Health applicants come) is an area for potential further work.
6	The findings of differential attainment do support, as stated in the discussion, 'review of the psychometric tests used' and 'additional support for candidates from disadvantaged groups' but it is inconsistent to argue for 'different thresholds set for different groups of candidates' at the same time as arguing that these assessments are not valid. It is difficult to see on what basis this could be done, and no evidence is provided that this could work effectively or be deemed acceptable to candidates or those selecting them.	We accept the point being made – that the findings do not necessitate such affirmative action, but we would highlight that the paragraph sets out the possibilities as “policy response options”. We had sought to cover a gamut of policy response actions. However, we have removed the clause about different thresholds. In our preceding paper,¹⁹ we outlined some of the potential responses to apparent differential attainment in the recruitment process more generally. In the context of these findings from the Assessment Centre stage specifically, the potential response options include a comprehensive review of the psychometric tests used to devise an assessment process and additional support for candidates from disadvantaged groups. or more affirmative action approaches that could see different thresholds set for different groups of candidates.
7	The authors state that 'observed differential attainment cannot be attributed to confounding alone: differential attainment was observed even within the UK medical graduate group. Instead, it appears that	We have now qualified this statement – see response (2).

	the system is unfairly discriminating against certain candidates.' This is at odds with the statement in the 'Strengths and limitations of the study' that 'Socioeconomic disadvantage and intersectionality are inherently complex and difficult to quantify; we cannot exclude the possibility of other potential confounders.'	
8	Some of the 'precise drivers' or 'possible explanations' such as test familiarity, are mentioned in the discussion but not accounted for in the analysis, but other known confounders such as educational background and specific learning difficulty are not mentioned. The statement in the abstract that 'psychometric tests demonstrate unwarranted and unacceptable differential attainment' is therefore not supported. Rather, differential attainment is unexplained by the factors accounted for in the analysis presented and how these were operationalised.	We have added the following statement to the paragraph cited, which highlights this as a deficit and potential challenge: Such putative explanatory variables are extremely difficult to measure or indeed analyse in a quantitative methodology. Thank you for this suggestion – the abstract is suitably amended: these psychometric tests demonstrate unexplained variation that suggests differential attainment
	Reviewer 2 comment (FP) Editor comment: BMJ Open does not consider relevance or novelty, points of particular concern to reviewer 2. While we do not want you to revise your research question or methods, we feel that some of the comments by reviewer 2 may inform your discussion.	Response
9	This paper addresses a very important research area regarding differential attainment (DA) in selection practices. It is a generally well-written paper with a well-executed analysis. However, my main reservation (as the authors acknowledge themselves), is that the material covered is a (relatively small) single specialty in a single recruitment round, based only in the UK. Whilst the findings will be of interest to policymakers involved in UK Public Health specialty recruitment (i.e., identified DA), I do not believe this paper currently tells us anything new regarding DA and selection methods/psychometric testing. Therefore, this paper in its current form is unlikely to be of sufficient interest to the broad readership of the journal/relevance internationally.	Thank you for the points raised - they are fair and relevant. We are particularly grateful for the extensive and detailed list of references – many of which we have now incorporated into the manuscript which we feel enhances the theoretical basis and context of this work. However, recognising the clarification made by the editor regarding the criteria for inclusion in BMJ Open, we contend that the findings are important and introduce potential questions that could be included in other specialties' recruitment processes.

10	This is not to say that the topic of DA and selection practices does not have relevance more generally, rather I'm not sure what this paper adds to what is already known regarding DA (and psychometric testing specifically). There are many existing large-scale studies identifying and exploring DA in selection and assessment (including meta-analytic studies within medical education e.g., Woolf et al, 2013 and more recent meta-analytic studies in the selection research and psychometric testing literature covering a broad range of occupational groups, e.g., Sackett, et al, 2022).	Public Health undertakes a recruitment process that is unique: it is both large (N>1000) and involves a heterogeneous pool of applicants (including those from outside the medical profession). Attempts to expand the process to other specialties is methodologically and analytically problematic. While we recognise the DA observed may not be necessarily novel for those with interests in psychometric testing, there is a broader point that is of interest to clinicians involved in recruitment.
11	Several existing studies offer a deeper analysis and insight into the issues regarding DA beyond this current paper. We have known for some while that DA exists in selection practices, the key challenge for researchers is to arrive at a better (causal) explanation as to why such differences exist, to inform more targeted interventions and policy initiatives to be of interest to the broader academic community. Although this paper is generally well-written, my advice would be for the authors to extend the current analysis to explore causal factors and psychometric test/selection method/selection process design interventions that have addressed the DA issue.	We whole-heartedly agree that the policy challenge is the principal issue – although that is outside the scope of this paper. We have added extensively to the theoretical context (see subsequent points and changes).
12	Originality - does the work add enough to what is already in the published literature? If so, what does it add? Please cite relevant references to support your comments on originality. Being aware of, and monitoring, DA in medical education and psychometric testing is a very important issue. The authors provide a more granular look into individual elements of the UK Public Health assessment process to identify potential sources of sub-group differences. However, this article does not add much beyond what is already present in current literature, especially within the selection literature (e.g. Bury et al., 2022; Fyfe et al., 2021; Patterson et al, 2016; Kelly et al., 2021; Sackett et al., 2021). There is a substantial evidence base which suggests that group differences are present across almost all selection methods (including psychometric tests).	The article is written as part of a commission from Health Education England and the Faculty of Public Health to investigate this issue for Public Health recruitment. While our previous paper (Bury et al 2022) explores the multi-stage process overall, this paper builds on the findings of that paper in three novel (albeit niche) ways:  i. Instrument by instrument analysis of the psychometric stage of testing ii. Includes analysis of additionally collected putative confounding variables. iii. We separately analysed international medical graduates.

	Examples include CVs (Derous & Ryan 2018), cognitive ability tests (Sackett et al., 2021; Ones et al., 2017) and interviews (Tridente et al., 2022). Although this paper does provide a granular look into individual elements of this specific and unique assessment process to identify DA there is insufficient material as it stands to demonstrate originality. In order to demonstrate originality I would encourage the authors to look at the broader literature as the starting point and then present data and analysis as case material which takes the debate to the next level in examining potential casual factors (to include an analysis of the selection criteria, scoring approaches and the interview etc, beyond the psychometric tests alone and follow-up interventions put in place that aim to reduce DA). A more holistic approach (which is where the literature in this area is heading) to include how DA has been addressed in the selection system would be of much broader interest in dealing with the complexity of the issues.	Furthermore, Public Health involves a much more heterogeneous pool of applicants than other medical specialties. While we agree that further investigation and discussion as suggested would be useful and welcome, they are not within the scope of the issue we were commissioned to explore.
13	Importance of the work to general readers - does this work matter to clinicians, researchers, policymakers, educators, or patients? Will it help our readers to make better decisions and, if so, how? Is a general medical journal the right place for it? Identifying the presence of DA in psychometric testing is of limited interest to researchers, since the paper does not make any alternative suggestions for methods of assessment, demonstrate what alternative interventions work (or not), such that practically it will not help readers to make changes or better decisions. The article may be better placed in a Public Health Journal as the topic is specific to Public Health and selection methods within that, which as the authors describe, is an “unusual specialty that recruits both medical and non-medical candidates”. Therefore, it may not be of interest to the wider readership as it is not directly transferable to other specialties or the use of these selection tools in other contexts. The paper is based on UK case material that may not be of huge relevance to an international audience, particularly those where their recruitment practices differ greatly. If the case is to be made regarding DA and psychometric testing more broadly, then drawing upon international research on this topic would help improve the importance of the work to the general readership.	We agree that upon exploration of the evidence base the presence of DA in psychometric testing is not novel. Yet while recognised by psychometricians, the awareness of these challenges in the broader recruitment space by clinicians over-seeing national recruitment processes is minimal. This commission was in response to that uncertainty, and on the back of several years of discussion around suspected confounders. We would therefore argue that these findings are of relevance and importance to clinicians and policy-makers – both within and outside the Public Health space. In most other international comparator countries with Public Health specialty training (or its equivalence), there is no entry point for those from backgrounds other than medicine. Accordingly, it is problematic to extrapolate internationally.

14	Is the research question clearly defined and appropriately answered? The research questions are generally well-defined although they could be stated more clearly. The authors state that following the findings from their previous study they are looking at how cultural factors could influence SJT scores, and language could influence scores of the other elements of the assessment centre. Cultural factors and what is included within that analysis would benefit from clearer definition.	Thank you for this point. We have sought to write this more clearly: Also of interest was whether the SJT specifically might be contributing to differential attainment by assuming cultural understanding of the British workplace: for example in assessing etiquette around punctuality and other culturally nuanced professional behaviours. Finally, we sought to assess whether having a first language other than English affects performance in the time-pressured WGCTAI assessment. We have also ensured consistency later on with the term ‘sociocultural’ – explained in the methods and later used in the discussion.
15	Overall design of study - appropriate and adequate to answer the research question? The authors explain their reasoning for only looking at the psychometric element of the Public Health recruitment process however, it is important to look at all aspects of the selection process together for this cohort in order to have an accurate picture of the contribution of each element to the DA they are observing in their data. The authors argue that their previous paper has identified a problem with the psychometric element of the process however, this analysis was run on previous cohorts (2018-2020), and we cannot assume that the results will match those from the 2021 cohort. Research suggests using multiple selection tools and using a compensatory method improves group differences outcomes (e.g., Stegers-Jager, 2017). I think the paper would be greatly improved if the authors included all stages of the recruitment process as more comprehensive information/measures in their analysis. This would greatly help the broader discussion, especially in exploring the causal factors at play.	We acknowledge this as a fair criticism of the paper. The complete process from the three preceding years is published in Bury et al 2021 (which is referenced). Ideally we would have liked to be able to use multiple years of entry for this analysis too. However, we were unable to do this because:  1. The additional sociocultural variables were only collected for the year that is presented in this proposed manuscript. Therefore we cannot extrapolate comparisons across the preceding years, except for acknowledging performance in the psychometric stage (56.8% passing) that is very similar to what we have published in Bury et al 2022 (56.5% for 2020). 2. The 2021 recruitment was complicated by the COVID-19 pandemic which meant that the previous hotel-based multi-component selection centre assessment had to be switched to an online interview based format. It is not possible (nor indeed desirable) to compare the different selection centre modalities directly.

		Accordingly we have added a further statement to the strengths and weaknesses section: The single-year nature of the dataset is also a limitation, although we note the proportion of candidates passing the AC in this year (56.8%) is very similar to the preceding year's cohort (56.5%).
16	Participants - adequately described, their conditions defined, inclusion and exclusion criteria described? How representative were the authors of patients whom this evidence might affect? The target sample has been clearly stated by the authors however, the inclusion criteria could be described with more detail. For example, the authors state that the dataset was partially redacted but do not clarify on what basis. Authors could consider restructuring this section to better the flow of the article and increase reader understanding. Furthermore, the authors may also want to consider/acknowledge whether the 2021 data may have been affected by the COVID-19 pandemic, which has been apparent in other recent selection research studies.	Thank you for highlighting this – we have now clarified the statement with: (to comply with the principle of data minimisation). We also believe that the restructured final paragraph of the introduction also strengthens the flow. Thank you for the query about the impact of COVID-19. This is an important point and we have now included it as a specific limitation: The single-year nature of the dataset is also a limitation, although we note the proportion of candidates passing the AC in this year (56.8%) is very similar to the preceding year's cohort (56.5%). It is not possible to infer what effect the COVID-19 pandemic might have had (with the Assessment Centre taking place in February 2021) although we note that most candidates completed the psychometric testing at home, whereas in previous years the testing had taken place in commercial test centres across the UK and internationally.
17	Methods - adequately described? Main outcome measure clear? Is the study fully reported in line with the appropriate reporting statement or checklist? Was the study ethical (this may go beyond simply whether the study was approved by an ethics committee or IRB)? The methods section and statistical analysis section is generally clear and coherent although a little vague in parts (for example, it is mentioned that the data was cleaned and coded, and it would be useful to provide	Thank you for the points. We recognise the statistical inferential limitation caused by the comparative small groups, and have referred to this in the limitations: Among the study's weaknesses is that ethnic groupings had to be aggregated to mitigate the analytical risk arising from small numbers. Accordingly, Type I and Type II errors cannot be

	an explanation as to how this was done). The research method and statistical analysis are appropriate to answer the research question and the outcome measure is clearly stated. There was some consideration for sample sizes when creating ethnic subcategories. The authors attempt to group ethnicities into categories that will allow for meaningful interpretation while also maintaining as much granularity. However, the sample sizes for some of the ethnic subgroups is still small, and there are large differences in size between the groups, leading the reader to question whether meaningful conclusions can be drawn from this.	ruled out. However, the consistency and strength of the findings around ethnicity and professional background suggest that a Type I error is highly unlikely in this context.
18	Results - answer the research question? Credible? Well presented? I am not fully familiar with the methodology deployed but the choice of analysis seems appropriate, and the results have been adequately presented and interpreted.	Thank you.
19	Interpretation and conclusions - warranted by and sufficiently derived from/focused on the data? Discussed in the light of previous evidence? Is the message clear? The authors of the article reach a largely appropriate conclusion and interpretations of the findings with a clear message that the current psychometric tests used in the selection process shows some evidence of DA and needs to be reviewed. There is limited reference to previous research as a way of backing up claims and interpretations of the findings in the discussion. The authors should also work on placing these findings into the wider context of DA in selection (as suggested above). The discussion does not currently sufficiently acknowledge the complexity of the issues in that DA is a complex, multi-faceted issue to which there is no single simple solution (see Woolf, 2020; Clapp & Gordon, 2021). The interpretation seems stops short as it assumes the group differences which are observed are equivalent to negative bias, which is not necessarily the case, which needs to be discussed (Davis et al., 2013; Reynolds, Altmann & Allen, 2021). The study has also been conducted on a cohort (2021) that was affected by Covid. Therefore, there are a number of potentially confounding variables that might have influenced the results which have not been considered such as test-	Thank you for the points made. We have sought to reference more of the evidence base (please see points below). We have also added a further sentence to the conclusion citing Clapp and Gordon 2021, and Woolf 2020. The drivers of differential attainment are complex, and policy responses even more so. We have included the Clapp and Gordon 2021 reference in respect of the need for holistic assessment. The section also references Reynolds and the CTBH. There is also a more fundamental limitation of quantitative approaches to 'measuring' diversity:³⁶ the quantitative paradigm is problematic in recognising intersectionality as well as other

taking environment and access to resources. Factors such as the weighting of the tests may also have had an impact on the group difference outcomes observed in practice (Lievens, Sackett & De Corte, 2021). Ensuring the person specifications and test specification is regularly reviewed and updated is also very important so that they remain relevant to the role, especially given changes to working in a world post-Covid. The authors stress the relevance of the results to policymakers and other specialties, however, make no attempt to suggest ways to mitigate the group differences that are observed or outline any practical actions that could be taken. Further discussion here would greatly improve the paper in the introduction the authors reference the UK Multi-specialty recruitment assessment (MSRA) and draw comparisons between the Public Health recruitment process and the MSRA. The MSRA is a very different selection test to those used in UK Public Health (which are cognitive ability tests and an SJT that assumes no clinical knowledge), as the MSRA is a test of procedural and clinical knowledge (see Lievens & Patterson, 2011). There also exists a substantial body of work exploring DA in the MSRA (e.g., see Patterson et al, 2018) to identify causal factors which need to be referenced. I think the authors would be better served to make more general comments about psychometric tests used for screening/short-listing purposes, to also be of interest to a broader audience.

contributing factors that motivate fairer selection. Some psychometricians have also challenged the assumption (the so-called Cultural Test Bias Hypothesis) that the observation of mean differences can be inferred as bias.³⁷ But in the absence of alternatives and the sheer volume of applicants and the differences observed, this analysis is at least a start.

A qualification relating to interpretation in the context of COVID-19 is made in point (16).

Our recommendations for further action include further investigation and points are made in the report and preceding paper (Bury 2022).

We have clarified the distinction between the PH process and the MSRA and inserted an additional reference to Lievens and Patterson 2011:

The use and weighting accorded to psychometric tests for postgraduate medical recruitment the UK varies. The Multi-Specialty Recruitment Assessment¹⁷ (MSRA) is a multi-instrument assessment of procedural and clinical knowledge that began in general practice and which has also been used in ophthalmology and obstetrics and gynaecology for several years.

We have also removed the reference to MSRA in the abstract:

~~*While other specialties should enhance their data collection to evaluate the impact of differential attainment on current selection processes, the roll-out of the psychometric Multi-Specialty Recruitment Assessment (MSRA) across further specialties in the UK should be urgently explored in respect of differential attainment.*~~

We have also stated and cited the paper Patterson et al 2018 in relation to the previously identified

		association between English fluency and SJT performance in general practice: We had also hypothesised that the use of SJTs in other medical application processes might have conferred an advantage to UK medical graduates; and language fluency has previously been associated with SJT performance in the recruitment process for general practice.²⁷ We have added the Sackett 2022 paper to the statement in the discussion about differential attainment in a wide range of psychometric tests – and not included it in the introduction due to the constraint on space, and the fact that the Sackett paper goes beyond healthcare into selection more generally.
20	References - up to date and relevant? Any glaring omissions? The references included are largely relevant and up to date. However, more research could be included as the authors currently do not consider important (recent and more broad ranging) references such as Sackett et al, 2022. Some of the latest research on the topic of DA in the last couple of years, particularly in a healthcare/assessment context, should also be included.	We have now updated the references to include many of the points made by the reviewer.
21	Abstract/summary/key messages/what this paper adds - reflect accurately what the paper says While the conclusions drawn lack some important considerations (as above), the abstract, summary and key messages are reflective of the content of the paper.	Thank you for this, and we note the removal of MSRA from the abstract per point (19).
	Reviewer 1 comment (AS)	Response

1 Thank you for asking me to review this paper. The inclusion of parental socioeconomic status and first language of candidates in the analysis is novel. The general conclusion that psychometric tests need to be investigated for differential attainment and the reasons for these explored is supported by the study and that an alternative valid and fair assessment should be considered is a fair implication.	Thank you for your comments – these are very helpful and we hope you will agree with us that the addition of these further limitations and citation of the papers mentioned improves the quality of the paper. We have provided a point-by-point response below.
2 The problem of unknown confounders is listed in the limitations but even known confounders are not accounted for (see below). The discussion then ignores this gap in the analysis and states that differential attainment is 'unfairly discriminating against certain candidates'.	We accept this point and have sought to qualify our conclusions. We have replaced the phrase “this study confirms” with “this study suggests” and we have also qualified later in the sentence that “likely” differential attainment was observed. In response to the citation in the reviewer response, we note that it was preceded with “it appears”: However, this study suggests that the observed differential attainment cannot be attributed to confounding alone: likely differential attainment was observed even within the UK medical graduate group. Instead, it appears that the system is unfairly discriminating against certain candidates. We have added the following into the first paragraph under “Strengths and weaknesses of the study”: Yet residual confounding cannot be ruled-out: the sociocultural variables used in the regression are imperfect, and it is very possible that other confounding variables were in-play.
3 Assertions in the paper, in particularly, that the results mean unfair discrimination against minority ethnic candidates, and the solutions presented, for example to set different thresholds for different groups of candidates are inconsistent with the analysis and other arguments presented in the paper.	See response to (6)
4 There are limitations which are not discussed. One potential confounder is specific learning difficulty which can affect performance, particularly where this is not declared or where reasonable accommodations are not provided (see Postgraduate Medical Journal	Thank you – this is a really helpful suggestion and we have added this to the “Unanswered questions and future research”:

	2018; 94:198–203. DOI:10.1136/postgradmedj-2017-135326.).	However, a range of additional variables may be useful. Neurodiversity was not included in the analysis although candidates declaring such conditions were provided with reasonable adjustments which included additional time and / or adjustments to the exam logistics. Previous work among trainees in general practice has observed higher prevalence of dyslexia among international medical graduates who had failed their first applied knowledge test (AKT).²⁹ Likewise, prior education or educational performance was also not included in the analysis. Previous has work has suggested possible intersectional associations with ethnicity in respect of prior education.³⁰ Capturing such data (while acknowledging the arguably wider range of educational backgrounds from which Public Health applicants come) is an area for potential further work.
5	Another important confounder, that is not included in the analysis, is prior education which is known to be linked to performance in undergraduate and postgraduate exams in medicine (BMC Med 2013;11:242. doi: 10.1186/1741-7015-11-242). This will be important for international medical graduates and but may be a factor even in UK graduates, whether medical or non-medical. As acknowledged by the authors, complex sociocultural factors affecting education may also intersect with ethnicity in ways that are not accounted for by a broad measure such as socioeconomic status.	However, a range of additional variables may be useful. Neurodiversity was not included in the analysis although candidates declaring such conditions were provided with reasonable adjustments which included additional time and / or adjustments to the exam logistics. Previous work among trainees in general practice has observed higher prevalence of dyslexia among international medical graduates who had failed their first applied knowledge test (AKT).²⁹ Likewise, prior education or educational performance was also not included in the analysis. Previous has work has suggested possible intersectional associations with ethnicity in respect of prior education.³⁰ Capturing such data (while acknowledging the arguably wider range of educational backgrounds from which Public Health applicants come) is an area for potential further work.
6	The findings of differential attainment do support, as stated in the discussion, 'review of the psychometric tests used' and 'additional support for candidates from disadvantaged groups' but it is inconsistent to argue for 'different thresholds set for different groups of candidates' at the same time as arguing that these assessments are not valid. It is difficult to see on what basis this could be done, and no evidence is provided that this could work effectively or be deemed acceptable to candidates or those selecting them.	We accept the point being made – that the findings do not necessitate such affirmative action, but we would highlight that the paragraph sets out the possibilities as “policy response options”. We had sought to cover a gamut of policy response actions. However, we have removed the clause about different thresholds. In our preceding paper,¹⁹ we outlined some of the potential responses to apparent differential attainment in the recruitment process more generally. In the context of these findings from the Assessment Centre stage specifically, the potential response options include a comprehensive review of the psychometric tests used to devise an assessment process and additional support for candidates from disadvantaged groups. or more affirmative action approaches that could see different thresholds set for different groups of candidates.
7	The authors state that 'observed differential attainment cannot be attributed to confounding alone: differential attainment was observed even within the UK medical graduate group. Instead, it appears that the system is unfairly discriminating against certain candidates.' This is at odds with the statement in the 'Strengths and limitations of the study' that 'Socioeconomic disadvantage and intersectionality are inherently	We have now qualified this statement – see response (2).

	complex and difficult to quantify; we cannot exclude the possibility of other potential confounders.'	
8	Some of the 'precise drivers' or 'possible explanations' such as test familiarity, are mentioned in the discussion but not accounted for in the analysis, but other known confounders such as educational background and specific learning difficulty are not mentioned. The statement in the abstract that 'psychometric tests demonstrate unwarranted and unacceptable differential attainment' is therefore not supported. Rather, differential attainment is unexplained by the factors accounted for in the analysis presented and how these were operationalised.	We have added the following statement to the paragraph cited, which highlights this as a deficit and potential challenge: Such putative explanatory variables are extremely difficult to measure or indeed analyse in a quantitative methodology. Thank you for this suggestion – the abstract is suitably amended: these psychometric tests demonstrate unexplained variation that suggests differential attainment
	Reviewer 2 comment (FP) Editor comment: BMJ Open does not consider relevance or novelty, points of particular concern to reviewer 2. While we do not want you to revise your research question or methods, we feel that some of the comments by reviewer 2 may inform your discussion.	Response
9	This paper addresses a very important research area regarding differential attainment (DA) in selection practices. It is a generally well-written paper with a well-executed analysis. However, my main reservation (as the authors acknowledge themselves), is that the material covered is a (relatively small) single specialty in a single recruitment round, based only in the UK. Whilst the findings will be of interest to policymakers involved in UK Public Health specialty recruitment (i.e., identified DA), I do not believe this paper currently tells us anything new regarding DA and selection methods/psychometric testing. Therefore, this paper in its current form is unlikely to be of sufficient interest to the broad readership of the journal/relevance internationally.	Thank you for the points raised - they are fair and relevant. We are particularly grateful for the extensive and detailed list of references – many of which we have now incorporated into the manuscript which we feel enhances the theoretical basis and context of this work. However, recognising the clarification made by the editor regarding the criteria for inclusion in BMJ Open, we contend that the findings are important and introduce potential questions that could be included in other specialties' recruitment processes.
10	This is not to say that the topic of DA and selection practices does not have relevance more generally, rather I'm not sure what this paper adds to what is	Public Health undertakes a recruitment process that is unique: it is both large (N>1000) and involves a heterogeneous pool of applicants

	already known regarding DA (and psychometric testing specifically). There are many existing large-scale studies identifying and exploring DA in selection and assessment (including meta-analytic studies within medical education e.g., Woolf et al, 2013 and more recent meta-analytic studies in the selection research and psychometric testing literature covering a broad range of occupational groups, e.g., Sackett, et al, 2022).	(including those from outside the medical profession). Attempts to expand the process to other specialties is methodologically and analytically problematic. While we recognise the DA observed may not be necessarily novel for those with interests in psychometric testing, there is a broader point that is of interest to clinicians involved in recruitment.
11	Several existing studies offer a deeper analysis and insight into the issues regarding DA beyond this current paper. We have known for some while that DA exists in selection practices, the key challenge for researchers is to arrive at a better (causal) explanation as to why such differences exist, to inform more targeted interventions and policy initiatives to be of interest to the broader academic community. Although this paper is generally well-written, my advice would be for the authors to extend the current analysis to explore causal factors and psychometric test/selection method/selection process design interventions that have addressed the DA issue.	We whole-heartedly agree that the policy challenge is the principal issue – although that is outside the scope of this paper. We have added extensively to the theoretical context (see subsequent points and changes).
12	Originality - does the work add enough to what is already in the published literature? If so, what does it add? Please cite relevant references to support your comments on originality. Being aware of, and monitoring, DA in medical education and psychometric testing is a very important issue. The authors provide a more granular look into individuals elements of the UK Public Health assessment process to identify potential sources of sub-group differences. However, this article does not add much beyond what is already present in current literature, especially within the selection literature (e.g. Bury et al., 2022; Fyfe et al., 2021; Patterson et al, 2016; Kelly et al., 2021; Sackett et al., 2021). There is a substantial evidence base which suggests that group differences are present across almost all selection methods (including psychometric tests). Examples include CVs (Derous & Ryan 2018), cognitive ability tests (Sackett et al., 2021; Ones et al., 2017) and interviews (Tridente et al., 2022). Although this paper does provide a granular look into individual elements of this specific and unique assessment	The article is written as part of a commission from Health Education England and the Faculty of Public Health to investigate this issue for Public Health recruitment. While our previous paper (Bury et al 2022) explores the multi-stage process overall, this paper builds on the findings of that paper in three novel (albeit niche) ways:  iv. Instrument by instrument analysis of the psychometric stage of testing v. Includes analysis of additionally collected putative confounding variables. vi. We separately analysed international medical graduates. Furthermore, Public Health involves a much more heterogeneous pool of applicants than other medical specialties.

	process to identify DA there is insufficient material as it stands to demonstrate originality. In order to demonstrate originality I would encourage the authors to look at the broader literature as the starting point and then present data and analysis as case material which takes the debate to the next level in examining potential casual factors (to include an analysis of the selection criteria, scoring approaches and the interview etc, beyond the psychometric tests alone and follow-up interventions put in place that aim to reduce DA). A more holistic approach (which is where the literature in this area is heading) to include how DA has been addressed in the selection system would be of much broader interest in dealing with the complexity of the issues.	While we agree that further investigation and discussion as suggested would be useful and welcome, they are not within the scope of the issue we were commissioned to explore.
13	Importance of the work to general readers - does this work matter to clinicians, researchers, policymakers, educators, or patients? Will it help our readers to make better decisions and, if so, how? Is a general medical journal the right place for it? Identifying the presence of DA in psychometric testing is of limited interest to researchers, since the paper does not make any alternative suggestions for methods of assessment, demonstrate what alternative interventions work (or not), such that practically it will not help readers to make changes or better decisions. The article may be better placed in a Public Health Journal as the topic is specific to Public Health and selection methods within that, which as the authors describe, is an “unusual specialty that recruits both medical and non-medical candidates”. Therefore, it may not be of interest to the wider readership as it is not directly transferable to other specialties or the use of these selection tools in other contexts. The paper is based on UK case material that may not be of huge relevance to an international audience, particularly those where their recruitment practices differ greatly. If the case is to be made regarding DA and psychometric testing more broadly, then drawing upon international research on this topic would help improve the importance of the work to the general readership.	We agree that upon exploration of the evidence base the presence of DA in psychometric testing is not novel. Yet while recognised by psychometricians, the awareness of these challenges in the broader recruitment space by clinicians over-seeing national recruitment processes is minimal. This commission was in response to that uncertainty, and on the back of several years of discussion around suspected confounders. We would therefore argue that these findings are of relevance and importance to clinicians and policy-makers – both within and outside the Public Health space. In most other international comparator countries with Public Health specialty training (or its equivalence), there is no entry point for those from backgrounds other than medicine. Accordingly, it is problematic to extrapolate internationally.
14	Is the research question clearly defined and appropriately answered?	Thank you for this point. We have sought to write this more clearly:

	The research questions are generally well-defined although they could be stated more clearly. The authors state that following the findings from their previous study they are looking at how cultural factors could influence SJT scores, and language could influence scores of the other elements of the assessment centre. Cultural factors and what is included within that analysis would benefit from clearer definition.	Also of interest was whether the SJT specifically might be contributing to differential attainment by assuming cultural understanding of the British workplace: for example in assessing etiquette around punctuality and other culturally nuanced professional behaviours. Finally, we sought to assess whether having a first language other than English affects performance in the time-pressured WGCTAI assessment. We have also ensured consistency later on with the term 'sociocultural' – explained in the methods and later used in the discussion.
15	Overall design of study - appropriate and adequate to answer the research question? The authors explain their reasoning for only looking at the psychometric element of the Public Health recruitment process however, it is important to look at all aspects of the selection process together for this cohort in order to have an accurate picture of the contribution of each element to the DA they are observing in their data. The authors argue that their previous paper has identified a problem with the psychometric element of the process however, this analysis was run on previous cohorts (2018-2020), and we cannot assume that the results will match those from the 2021 cohort. Research suggests using multiple selection tools and using a compensatory method improves group differences outcomes (e.g., Stegers-Jager, 2017). I think the paper would be greatly improved if the authors included all stages of the recruitment process as more comprehensive information/measures in their analysis. This would greatly help the broader discussion, especially in exploring the causal factors at play.	We acknowledge this as a fair criticism of the paper. The complete process from the three preceding years is published in Bury et al 2021 (which is referenced). Ideally we would have liked to be able to use multiple years of entry for this analysis too. However, we were unable to do this because:  3. The additional sociocultural variables were only collected for the year that is presented in this proposed manuscript. Therefore we cannot extrapolate comparisons across the preceding years, except for acknowledging performance in the psychometric stage (56.8% passing) that is very similar to what we have published in Bury et al 2022 (56.5% for 2020). 4. The 2021 recruitment was complicated by the COVID-19 pandemic which meant that the previous hotel-based multi-component selection centre assessment had to be switched to an online interview based format. It is not possible (nor indeed desirable) to compare the different selection centre modalities directly. Accordingly we have added a further statement to the strengths and weaknesses section:

		The single-year nature of the dataset is also a limitation, although we note the proportion of candidates passing the AC in this year (56.8%) is very similar to the preceding year's cohort (56.5%).
16	Participants - adequately described, their conditions defined, inclusion and exclusion criteria described? How representative were the authors of patients whom this evidence might affect? The target sample has been clearly stated by the authors however, the inclusion criteria could be described with more detail. For example, the authors state that the dataset was partially redacted but do not clarify on what basis. Authors could consider restructuring this section to better the flow of the article and increase reader understanding. Furthermore, the authors may also want to consider/acknowledge whether the 2021 data may have been affected by the COVID-19 pandemic, which has been apparent in other recent selection research studies.	Thank you for highlighting this – we have now clarified the statement with: (to comply with the principle of data minimisation). We also believe that the restructured final paragraph of the introduction also strengthens the flow. Thank you for the query about the impact of COVID-19. This is an important point and we have now included it as a specific limitation: The single-year nature of the dataset is also a limitation, although we note the proportion of candidates passing the AC in this year (56.8%) is very similar to the preceding year's cohort (56.5%). It is not possible to infer what effect the COVID-19 pandemic might have had (with the Assessment Centre taking place in February 2021) although we note that most candidates completed the psychometric testing at home, whereas in previous years the testing had taken place in commercial test centres across the UK and internationally.
17	Methods - adequately described? Main outcome measure clear? Is the study fully reported in line with the appropriate reporting statement or checklist? Was the study ethical (this may go beyond simply whether the study was approved by an ethics committee or IRB)? The methods section and statistical analysis section is generally clear and coherent although a little vague in parts (for example, it is mentioned that the data was cleaned and coded, and it would be useful to provide an explanation as to how this was done). The research method and statistical analysis are appropriate to answer the research question and the outcome measure is clearly stated. There was some	Thank you for the points. We recognise the statistical inferential limitation caused by the comparative small groups, and have referred to this in the limitations: Among the study's weaknesses is that ethnic groupings had to be aggregated to mitigate the analytical risk arising from small numbers. Accordingly, Type I and Type II errors cannot be ruled out. However, the consistency and strength of the findings around ethnicity and professional background suggest that a Type I error is highly unlikely in this context.

	consideration for sample sizes when creating ethnic subcategories. The authors attempt to group ethnicities into categories that will allow for meaningful interpretation while also maintaining as much granularity. However, the sample sizes for some of the ethnic subgroups is still small, and there are large differences in size between the groups, leading the reader to question whether meaningful conclusions can be drawn from this.	
18	Results - answer the research question? Credible? Well presented? I am not fully familiar with the methodology deployed but the choice of analysis seems appropriate, and the results have been adequately presented and interpreted.	Thank you.
19	Interpretation and conclusions - warranted by and sufficiently derived from/focused on the data? Discussed in the light of previous evidence? Is the message clear? The authors of the article reach a largely appropriate conclusion and interpretations of the findings with a clear message that the current psychometric tests used in the selection process shows some evidence of DA and needs to be reviewed. There is limited reference to previous research as a way of backing up claims and interpretations of the findings in the discussion. The authors should also work on placing these findings into the wider context of DA in selection (as suggested above). The discussion does not currently sufficiently acknowledge the complexity of the issues in that DA is a complex, multi-faceted issue to which there is no single simple solution (see Woolf, 2020; Clapp & Gordon, 2021). The interpretation seems stops short as it assumes the group differences which are observed are equivalent to negative bias, which is not necessarily the case, which needs to be discussed (Davis et al., 2013; Reynolds, Altmann & Allen, 2021). The study has also been conducted on a cohort (2021) that was affected by Covid. Therefore, there are a number of potentially confounding variables that might have influenced the results which have not been considered such as test-taking environment and access to resources. Factors such as the weighting of the tests may also have had an impact on the group difference outcomes observed in practice (Lievens, Sackett & De Corte, 2021).	Thank you for the points made. We have sought to reference more of the evidence base (please see points below). We have also added a further sentence to the conclusion citing Clapp and Gordon 2021, and Woolf 2020. The drivers of differential attainment are complex, and policy responses even more so. We have included the Clapp and Gordon 2021 reference in respect of the need for holistic assessment. The section also references Reynolds and the CTBH. There is also a more fundamental limitation of quantitative approaches to 'measuring' diversity:³⁶ the quantitative paradigm is problematic in recognising intersectionality as well as other contributing factors that motivate fairer selection. Some psychometricians have also challenged the assumption (the so-called Cultural Test Bias Hypothesis) that the observation of mean

Ensuring the person specifications and test specification is regularly reviewed and updated is also very important so that they remain relevant to the role, especially given changes to working in a world post-Covid. The authors stress the relevance of the results to policymakers and other specialties, however, make no attempt to suggest ways to mitigate the group differences that are observed or outline any practical actions that could be taken. Further discussion here would greatly improve the paper in the introduction the authors reference the UK Multi-specialty recruitment assessment (MSRA) and draw comparisons between the Public Health recruitment process and the MSRA. The MSRA is a very different selection test to those used in UK Public Health (which are cognitive ability tests and an SJT that assumes no clinical knowledge), as the MSRA is a test of procedural and clinical knowledge (see Lievens & Patterson, 2011). There also exists a substantial body of work exploring DA in the MSRA (e.g., see Patterson et al, 2018) to identify causal factors which need to be referenced. I think the authors would be better served to make more general comments about psychometric tests used for screening/short-listing purposes, to also be of interest to a broader audience.

differences can be inferred as bias.³⁷ But in the absence of alternatives and the sheer volume of applicants and the differences observed, this analysis is at least a start.

A qualification relating to interpretation in the context of COVID-19 is made in point (16).

Our recommendations for further action include further investigation and points are made in the report and preceding paper (Bury 2022).

We have clarified the distinction between the PH process and the MSRA and inserted an additional reference to Lievens and Patterson 2011:

The use and weighting accorded to psychometric tests for postgraduate medical recruitment the UK varies. The Multi-Specialty Recruitment Assessment¹⁷ (MSRA) is a multi-instrument assessment of procedural and clinical knowledge that began in general practice and which has also been used in ophthalmology and obstetrics and gynaecology for several years.

We have also removed the reference to MSRA in the abstract:

~~*While other specialties should enhance their data collection to evaluate the impact of differential attainment on current selection processes, the roll-out of the psychometric Multi-Specialty Recruitment Assessment (MSRA) across further specialties in the UK should be urgently explored in respect of differential attainment.*~~

We have also stated and cited the paper Patterson et al 2018 in relation to the previously identified association between English fluency and SJT performance in general practice:

		We had also hypothesised that the use of SJTs in other medical application processes might have conferred an advantage to UK medical graduates; and language fluency has previously been associated with SJT performance in the recruitment process for general practice.²⁷ We have added the Sackett 2022 paper to the statement in the discussion about differential attainment in a wide range of psychometric tests – and not included it in the introduction due to the constraint on space, and the fact that the Sackett paper goes beyond healthcare into selection more generally.
20	References - up to date and relevant? Any glaring omissions? The references included are largely relevant and up to date. However, more research could be included as the authors currently do not consider important (recent and more broad ranging) references such as Sackett et al, 2022. Some of the latest research on the topic of DA in the last couple of years, particularly in a healthcare/assessment context, should also be included.	We have now updated the references to include many of the points made by the reviewer.
21	Abstract/summary/key messages/what this paper adds - reflect accurately what the paper says While the conclusions drawn lack some important considerations (as above), the abstract, summary and key messages are reflective of the content of the paper.	Thank you for this, and we note the removal of MSRA from the abstract per point (19).

- Fran Bury, Richard J Pinder, & Richard Pinder. (2022). Differential attainment in public health specialty training recruitment in the United Kingdom: an observational analysis of applicants from 2018 to 2020. *Journal of Public Health*. <https://doi.org/10.1093/pubmed/fdac122>
- Clapp, J. T., & Gordon, E. K. (2021). Selecting trainees: Too much focus on predictive metrics, not enough on holistic review. *Medical Education*, 56(2), 139–141. <https://doi.org/10.1111/medu.14704>
- Davis, D., Dorsey, J. K., Franks, R. D., Sackett, P. R., Searcy, C. A., & Zhao, X. (2013). Do Racial and Ethnic Group Differences in Performance on the MCAT Exam Reflect Test Bias? *Academic Medicine*, 88(5), 593–602. <https://doi.org/10.1097/acm.0b013e318286803a>
- Derous, E., & Ryan, A. M. (2018). When your resume is (not) turning you down: Modelling ethnic bias in resume screening. *Human Resource Management Journal*, 29(2), 113–130. <https://doi.org/10.1111/1748-8583.12217>
- Fyfe, M., Horsburgh, J., Blitz, J., Chiavaroli, N., Kumar, S., & Cleland, J. (2021). The do's, don'ts and don't knows of redressing differential attainment related to race/ethnicity in medical schools. *Perspectives on Medical Education*, 11(1), 1–14. <https://doi.org/10.1007/s40037-021-00696-3>
- Kelly, L., & Sankaranarayanan, S. (2021). Differential attainment: how can we close the gap in paediatrics? *Archives of Disease in Childhood - Education & Practice Edition*, edpract-2020. <https://doi.org/10.1136/archdischild-2020-321066>
- Lievens, F., & Patterson, F. (2011). The validity and incremental validity of knowledge tests, low-fidelity simulations, and high-fidelity simulations for predicting job performance in advanced-level high-stakes selection. *Journal of Applied Psychology*, 96(5), 927–940. <https://doi.org/10.1037/a0023496>
- Lievens, F., Sackett, P. R., & De Corte, W. (2021). Weighting admission scores to balance predictiveness-diversity: The Pareto-optimization approach. *Medical Education*, 56(2), 151–158. <https://doi.org/10.1111/medu.14606>
- Ones, D. S. S. D. (2017, March 27). *Cognitive Ability | 11 | v2 | Measurement and Validity for Employee Se*. Taylor & Francis. Retrieved October 7, 2022, from <https://www.taylorfrancis.com/chapters/edit/10.4324/9781315690193-11/cognitive-ability-deniz-ones-stephan-dilchert-chockalingam-viswesvaran-jes%C3%BAs-salgado>
- Patterson, F., Zibarras, L., & Ashworth, V. (2016). Situational judgement tests in medical education and training: Research, theory and practice: AMEE Guide No. 100. *Medical Teacher*, 38(1), 3–17. <https://doi.org/10.3109/0142159x.2015.1072619>
- Patterson F, Tiffin P, Lopes S & Zibarras L. (2018). Unpacking the 'dark variance' of differential attainment in professional exams for overseas graduates. *Medical Education*, 52(7)., DOI:10.1111/medu.13605
- Reynolds, C.R., Altmann, R.A., Allen, D.N. (2021). The Problem of Bias in Psychological Assessment. In: *Mastering Modern Psychological Testing*. Springer, Cham. https://doi.org/10.1007/978-3-030-59455-8_15
- Stegers-Jager, K. M. (2017b). Lessons learned from 15 years of non-grades-based selection for medical school. *Medical Education*, 52(1), 86–95. <https://doi.org/10.1111/medu.13462>
- Sackett, P et al (2022). Revisiting meta-analytic estimates of validity in personnel selection: Addressing systematic overcorrection for restriction of range. *Journal of Applied Psychology*, 107(11):2040-2068. doi: 10.1037/apl0000994.

Tridente, A., Parry-Jones, J., Chandrashekaraiyah, S., & Bryden, D. (2022). Differential attainment and recruitment to Intensive Care Medicine Training in the UK, 2018–2020. *BMC Medical Education*, 22(1). <https://doi.org/10.1186/s12909-022-03732-w>

Woolf, K. (2020). Differential attainment in medical education and training. *BMJ*, m339. <https://doi.org/10.1136/bmj.m339>